# Occurrence and Risk Factors of Relapse Activity after Vaccination against COVID-19 in People with Multiple Sclerosis: 1-Year Follow-Up Results from a Nationwide Longitudinal Observational Study

**DOI:** 10.3390/vaccines11121859

**Published:** 2023-12-16

**Authors:** Firas Fneish, Niklas Frahm, Melanie Peters, David Ellenberger, Judith Haas, Micha Löbermann, Dieter Pöhlau, Anna-Lena Röper, Sarah Schilling, Alexander Stahmann, Herbert Temmes, Friedemann Paul, Uwe K. Zettl

**Affiliations:** 1MS Forschungs- und Projektentwicklungs-gGmbH (MS Research and Project Development gGmbH [MSFP]), German MS Registry, 30171 Hannover, Germany; fneish@msregister.de (F.F.); peters@msregister.de (M.P.); ellenberger@msregister.de (D.E.); roeper@msregister.de (A.-L.R.); schilling@msregister.de (S.S.); stahmann@msregister.de (A.S.); 2Neuroimmunological Section, Department of Neurology, University Medical Center of Rostock, 18147 Rostock, Germany; uwe.zettl@med.uni-rostock.de; 3Gesellschaft für Versorgungsforschung mbH (Society for Health Care Research [GfV]), German MS Registry, 30171 Hannover, Germany; 4Deutsche Multiple Sklerose Gesellschaft, Bundesverband e.V. (German MS Society Federal Association [DMSG]), 30171 Hannover, Germany; haas-heide@gmx.de (J.H.); dieter.poehlau@kamillus-klinik.de (D.P.); temmes@dmsg.de (H.T.); 5Department of Tropical Medicine, Infectious Diseases and Nephrology, University Medical Center of Rostock, 18057 Rostock, Germany; micha.loebermann@med.uni-rostock.de; 6Experimental and Clinical Research Center, Joint Cooperation between the Max Delbrück Center for Molecular Medicine in the Helmholtz Association, The Charité Medical Faculty, Campus Berlin-Buch, 13125 Berlin, Germany; friedemann.paul@charite.de; 7Department of Neurology, Charité—Universitätsmedizin, 10117 Berlin, Germany; 8NeuroCure Clinical Research Center, Charité—Universitätsmedizin, 10117 Berlin, Germany

**Keywords:** SARS-CoV-2, vaccination, multiple sclerosis, relapse, side effects

## Abstract

Several studies reported post-SARS-CoV-2-vaccination (PV) symptoms. Even people with multiple sclerosis (PwMS) have concerns about disease activity following the SARS-CoV-2 vaccination. We aimed to determine the proportion of PwMS with PV relapses, the PV annualized relapse rate (ARR), the time from vaccination to subsequent relapses, and identify sociodemographic/clinical risk factors for PV relapses. PwMS were surveyed several times at baseline and four follow-ups as part of a longitudinal observational study regarding the safety and tolerability of the SARS-CoV-2 vaccination. The inclusion criteria for this analysis were age ≥18 years, ≥1 SARS-CoV-2 vaccination, and ≥1-year observation period since initial vaccination. Of 2466 PwMS, 13.8% reported PV relapses (mostly after second [N = 147] or booster vaccination [N = 145]) at a median of 8.0 (first/third quantile: 3.55/18.1) weeks PV, with the shortest period following initial vaccination (3.95 weeks). The ARR was 0.153 (95% confidence interval: 0.138–0.168), with a median observation period since initial vaccination of 1.2 years. Risk factors for PV relapses were younger age, female gender, moderate-severe disability levels, concurrent autoimmune diseases, relapsing-remitting MS courses, no DMT, and relapses within the year prior to the first vaccination. Patients’ health conditions before/during initial vaccination may play a more important role in PV relapse occurrence than vaccination per se.

## 1. Introduction

Severe acute respiratory syndrome coronavirus 2 (SARS-CoV-2) has rapidly spread throughout the world since 2020 [1]. Approximately 770 million people have been infected to date, while around 7 million people have died while infected [2]. The development and approval of novel vaccines (for example, tozinameran, elasomeran, AZD1222, Ad26.COV2.S, and NVX-Co2373 [3,4,5]) were expected to mitigate the pandemic to some degree. Consequently, vaccine development became the focus of scientific and public attention [3]. However, the relatively rapid development and approval of the novel vaccines were also met with skepticism [6]. One point of hesitancy was the occurrence of, for example, severe or life-threatening post-vaccination side effects with some of these vaccines [7,8]. People’s uncertainty about vaccination may have been heightened by the difference between widespread SARS-CoV-2 vaccination campaigns and the skepticism or changes in national and international vaccination recommendations during the SARS-CoV-2 pandemic. This also applies to people with chronic autoimmune diseases, such as multiple sclerosis (MS). This disease is the most prevalent neuroimmunological illness of the central nervous system and is commonly diagnosed in young adults [9,10]. Globally, there are approximately 2.8 million people with MS (PwMS), with a majority being women [11,12,13]. The disease presents a wide range of symptoms, such as fatigue, cognitive impairment, visual disturbances, gait impairment, bladder symptoms, and genital disorders [9,10].

It is precisely these PwMS who are especially prone to infectious diseases due to the disabling disease itself and the treatment with disease-modifying therapies (DMTs), resulting in a higher risk of mortality [14,15]. Thus, vaccination is the primary preventive strategy for avoiding infections in PwMS [16,17]. Studies have provided evidence for the beneficial effects of standard vaccination (e.g., against mumps, measles, rubella, or tetanus) in PwMS but have not found a correlation between the onset or worsening of MS and established vaccines [17,18]. An exception that has been widely known is the yellow fever vaccine, an attenuated vaccine that may cause disease activity in the form of relapses [19]. Nevertheless, there have been concerns among PwMS and treating physicians regarding the new SARS-CoV-2 vaccines, particularly fears of MS exacerbation due to possible bystander activation by vector-based vaccines or cross-reactivity by mRNA-based vaccines [20,21,22].

For this reason, we initiated data collection on 3 May 2021, as part of a nationwide longitudinal observational study investigating the safety and tolerability of the SARS-CoV-2 vaccination in PwMS in Germany. Prior analyses have yielded results regarding vaccine reactions [23] and short-term relapse activity (median observation periods since the first vaccination of 2 to 4 months) following SARS-CoV-2 vaccination [24] in over 2000 PwMS. With the current analysis, we are able to present, for the first time, data on post-vaccination disease activity with an observation period of ≥1 year since the first SARS-CoV-2 vaccination. This study aims to analyze the occurrence of post-vaccination relapses among PwMS (proportion of patients with post-vaccination relapses, annualized relapse rate [ARR] within the year following the first vaccination, time from vaccination to the subsequent relapse) and identify sociodemographic and clinical risk factors for post-vaccination relapse activity.

## 2. Materials and Methods

This prospective, non-interventional, observational study is based on a longitudinal online survey on the safety and tolerability of VACC in PwMS in Germany, consisting of five surveys. Patient-reported data had been collected via online questionnaires on the website of the German MS Registry. The German MS Society (DMSG) did recruitment via newsletters and social media posts. The questionnaires were collaboratively developed by the German MS Registry and experts from its scientific advisory board and the DMSG in consultation with the United Kingdom MS Registry and the MS Data Alliance. After development, several rounds of testing were conducted by employees of the MS Registry, DMSG, and PwMS. Patients of at least 18 years of age with a diagnosis of MS [25] and at least one vaccination against SARS-CoV-2 were eligible to participate in the baseline survey on 3 May 2021 after they provided informed consent. The participants of the baseline survey were invited to complete the first follow-up, the second follow-up, the booster survey, and the third follow-up after they should have received their second vaccination (according to the recommendations of the German Standing Committee on Vaccination [STIKO] [26]), three months after the second vaccination, after they should have received their first booster vaccination (usually after two vaccine doses), and approximately one year after the first vaccination, respectively.

Data collection at baseline included sociodemographic details (age, gender), clinical-neurological data on MS (e.g., MS course type, DMT status, presence of other autoimmune comorbidities [like psoriasis, autoimmune thyroid diseases, rheumatism, etc.] and degree of disability, measured by patient-determined disease steps [PDDS] [27]), and information on vaccination as well as disease activity (e.g., type of vaccine administered, date of the last relapse prior to the first vaccination, number of relapses after any vaccination). Detailed information on the study design and data collection, including the baseline and the first two follow-up questionnaires, has been published in two previous articles by Frahm et al. [23,24]. The current analysis is based on the data from the third follow-up survey, which also contained detailed questions regarding the occurrence (number and date) of relapses within ≥1 year following the first vaccination, as shown in Appendix A. We explicitly asked about relapses diagnosed by a physician (including the date of diagnosis). Only patients who provided data on post-vaccination relapses during the third follow-up were included in this analysis, regardless of whether a post-vaccination relapse occurred or was absent. PwMS who did not provide any information on the occurrence or absence of post-vaccination relapses were excluded.

### Statistics

The data collection for the third follow-up ended on 30 March 2023. Median values, including 25% and 75% quantiles (Q25 and Q75, respectively), and percentages were used to show the sociodemographic and clinical-neurological composition of the study population. Moreover, the proportion of PwMS reporting ≥1 post-vaccination relapse and the time to post-vaccination relapses were calculated. Comparisons of patients with and without post-vaccination relapses were conducted using the chi-square test, Fisher’s exact test, and Mann–Whitney U test, with the significance level set at α = 0.05. Non-parametric tests, such as the Mann–Whitney U test, were preferentially utilized in our analysis due to their robustness in situations where parametric assumptions, including normality, are not met. It is important to note that parametric tests assume specific properties of the population distribution, such as normality and homogeneity of variances, which may not always hold in real-world scenarios. This is particularly pertinent when working with large sample sizes [28,29,30]. Additionally, non-parametric tests enable hypothesis testing and group comparisons without assuming specific parameters of the population distribution, making them particularly advantageous when dealing with real-world data that may not align with theoretical expectations. Fisher’s exact test was chosen as an alternative for handling categorical endpoints when the assumptions of the chi-square test are not met. Various studies have explored and evaluated the appropriateness of different tests, including Fisher’s exact test [31,32,33,34,35]. Nevertheless, we chose to utilize Fisher’s exact test due to the tendency of the chi-square test with Yates correction to be conservative in controlling for type I error. Some studies suggest that the Fisher exact test is overly conservative with large sample sizes, but it remains more appropriate when one of the cell frequencies is less than five. The following periods were examined for the occurrence of post-vaccination relapse: from the first to the next SARS-CoV-2 vaccination (P1), from the second to the next vaccination (P2), and from the first booster vaccination to the end of observation (P3). If there was no further vaccination after the first or second vaccination, P1 and P2 were considered until the end of the observation period for the respective PwMS. ARRs from the first vaccination for the occurrence of post-vaccination relapses, including 95% confidence intervals (CIs), were calculated for the study population, stratified by age, gender, MS course type, disability level, DMT status at baseline, presence/absence of autoimmune comorbidities at baseline, and presence/absence of relapses within the year prior to the first vaccination. Univariable and multivariable negative binomial models were used to calculate risk ratios (RRs) and 95% CIs with observation time as an offset for the occurrence of post-vaccination relapses among PwMS, stratified by age, gender, degree of disability, DMT status at baseline, presence/absence of autoimmune comorbidities at baseline, and presence/absence of relapses within the year prior to the first vaccination. Statistical analyses, data transformation, and the generation of figures were performed using R 4.0 (The R Foundation for Statistical Computing, Vienna, Austria; packages used: glm, comparegroups, survival, survmin, alluvial, finalfit) and Microsoft Excel v2202 (Microsoft Corporation, Redmond, WA, USA).

## 3. Results

A total of 2466 PwMS were included in this study, while seven participants of the third follow-up survey were excluded due to missing data on post-vaccination relapse activity (Figure 1). The median observation period since the first vaccination was 1.2 (Q25, Q75: 1.1, 1.25) years. The study population is characterized by 78.6% female patients, a median age at baseline survey of 46.9 years, 74.1% RRMS patients, 52.3% PwMS with a mild level of disability (measured by PDDS), 21.5% patients with coincident autoimmune diseases at baseline, DMT in 73.2% of PwMS, relapse activity in the year prior to the first vaccination in 20.0% of patients, and a median time from the last pre-vaccination relapse to the first vaccination of 3.2 years (Table 1). The majority of included patients reported no relapse activity after any vaccination (86.2%), while 341 patients indicated experiencing ≥1 post-vaccination relapse (13.8%). In total, 434 post-vaccination relapses were reported. Patients with post-vaccination relapses showed higher proportions of women, RRMS, moderate disability degree, coincident autoimmune diseases at baseline, and relapse activity within the year prior to the first vaccination compared with patients without post-vaccination relapses. In addition, PwMS with post-vaccination relapses were median younger, less often treated with DMT, and had a shorter time from the last pre-vaccination relapse to the first vaccination (Table 1). The sociodemographic and clinical profiles of the patients (presented in Appendix A) were stratified according to the occurrence of post-vaccination relapses, age, and gender. Female PwMS in younger age groups experienced relapses more frequently than male patients in the same age range (18–30 years: 21.1% vs. 7.7%; 31–40 years: 21.2% vs. 11.7%).

Regarding the vaccination scheme, tozinameran was the most frequently administered vaccine regardless of the time of vaccination (first vaccination: 78.3%, second vaccination: 84.8%, first booster vaccination: 66.2%); see Table 2. At the time of the first vaccination, the proportion of patients receiving elasomeran was lower than at the time of the second vaccination and the first booster vaccination (9.8% vs. 11.9% vs. 33.2%). There were no significant differences in vaccine distribution at the time of the first vaccination, second vaccination, or first booster vaccination between patients with and without post-vaccination relapses (*p* ≥ 0.225). The detailed description of the vaccination schedules used, including the switch of vaccines from the first vaccination to the first booster vaccination, is shown in Appendix A, also for patients with and without post-vaccination relapses.

In the investigation period P1, 91 PwMS patients reported post-vaccination relapses, while 147 and 145 patients reported post-vaccination relapses in P2 and P3, respectively (patients may have experienced relapses in several periods). For the entire study population, relapses occurred at a median of 8.0 (Q25, Q75: 3.55, 18.1) weeks after any vaccination. The median time to relapse was the shortest after the first vaccination (3.95 [1.7, 5.6] weeks) and increased with subsequent vaccinations (second vaccination: 9.5 [3.9, 19.7] weeks; first booster vaccination: 13.5 [6.6, 20.3] weeks). The ARR for the total cohort was 0.153 (95% CI: 0.139–0.168). Furthermore, patients with an age ≤40 years, female gender, RRMS course, moderate disability level, no DMT at baseline, relapses within the year prior to the first vaccination, and autoimmune comorbidities had higher ARRs than PwMS with an age >40 years, male gender, SPMS/PPMS/undefined MS course, mild or severe disability level, DMT at baseline, no relapses within the year prior to the first vaccination, and no autoimmune comorbidities at baseline, respectively (see Figure 2). The majority of the 341 patients with post-vaccination relapses were treated (64.5%). This involved the use of glucocorticosteroids in 199 PwMS, immunoadsorption in 16 PwMS, and plasmapheresis in 12 PwMS. No treatment after a previous relapse was reported by 95 patients.

In the univariable negative binomial model (Appendix A), female gender (RR = 1.70 [95% CI: 1.29–2.25], *p* = 0.002; reference: male), moderate disability level (RR = 1.37 [1.10–1.71], *p* = 0.019; reference: mild), no DMT at baseline (RR = 1.55 [1.24–1.93], *p* = 0.001; reference: DMT), relapses within the year before the first vaccination (RR = 2.81 [2.21–3.57], *p* < 0.001; reference: no relapses), and autoimmune comorbidities at baseline (RR = 1.44 [1.14–1.83], *p* = 0.010; reference: no autoimmune comorbidity) were significantly associated with a higher risk of post-vaccination relapses. Higher age (41–50 years: RR = 0.52 [0.37–0.74], *p* = 0.002; 51–60 years: RR = 0.42 [0.29–0.60], *p* < 0.001; >60 years: RR = 0.48 [0.30–0.75], *p* = 0.008; reference: 18–30 years, respectively) and SPMS course (RR = 0.49 [0.36–0.68], *p* < 0.001; references: RRMS) were associated with a lower post-vaccination relapse risk. In the multivariable model, moderate (RR = 1.97 [1.54–2.52], *p* < 0.001; reference: mild) to severe (RR = 2.50 [1.56–3.99], *p* = 0.001; reference: mild) degree of disability, no DMT at baseline (RR = 1.79 [1.39–2.31], *p* < 0.001) and relapse activity within the year prior to the first vaccination (RR = 2.35 [1.86–2.98], *p* < 0.001, reference: no relapse) were identified as risk factors for the occurrence of post-vaccination relapses, whereas an age between 41 and 60 years (41–50 years: RR = 0.58 [0.40–0.84], *p* = 0.014; 51–60 years: RR = 0.52 [0.35–0.77], *p* = 0.007; reference: 18–30 years, respectively) and undefined (RR = 0.34 [0.13–0.77], *p* = 0.047, reference: RRMS) as well as SPMS course (RR = 0.41 [0.26–0.65], *p* = 0.002, reference: RRMS) were significantly associated with a lower risk of post-vaccination relapses (Figure 3).

## 4. Discussion

In the domain of SARS-CoV-2 vaccinations, the designated schedules and intervals for vaccination emerge from a careful consideration of several factors. Scientific insights, clinical trial data, vaccine availability, and epidemiological considerations guide the selection process. Examples of vaccination schedules include homologous strategies, which involve using the same types of vaccines for both doses, as well as heterologous approaches that involve different vaccine types [36]. In addition, the length of time between vaccine doses varies depending on the country [37]—longer intervals are preferred in some countries to achieve wider primary immunization, while shorter intervals are favored elsewhere for a more rapid onset of protection. To maintain vaccine protection, some countries also offer booster shots or supplementary doses [37]. Decisions on creating vaccination schedules are constantly evolving and influenced by new scientific discoveries, virus variations, and the supply of vaccines [38]. The variety of vaccination methods indicates endeavors to guarantee optimal effectiveness and safety in diverse epidemiological settings. However, safety concerns surrounding SARS-CoV-2 vaccines emerged during their developmental stages. There have been debates among physicians and patients, particularly regarding chronically ill individuals like PwMS, on the benefit-risk assessment of the newly developed vaccines. This update to our prospective, non-interventional study on the safety and tolerability of SARS-CoV-2 vaccines in PwMS expands upon existing scientific evidence by examining post-vaccination relapse activity among a substantial cohort of PwMS (N = 2466) over a median period of greater than one year since the initial vaccination.

Through a series of online surveys of PwMS following their initial vaccination against SARS-CoV-2, we were able to consistently gather and analyze data on relapse activity in the year following the first vaccination. The initial interim study focused on the prevalence of immediate vaccine reactions [23]. The study cohort from Germany (N = 2346) was observed for 2 months post first vaccination and compared with a cohort of 3796 PwMS from the UK, which was observed for 6 months. The primary outcomes of this first interim analysis suggested that immediate side effects such as fatigue, headache, and pain (at the injection site) were most prevalent after vaccination, with women having a significantly higher risk of experiencing these vaccine reactions. These findings align with other studies conducted on PwMS [8,39,40,41,42,43]. In terms of MS-specific change in disease progression after vaccination, 19% of the German cohort experienced self-reported worsening or new onset of MS symptoms, particularly fatigue and gait disturbance [23]. In the first interim analysis, relapse data were also available from the German PwMS: 141 of 2346 patients (6.0%) reported ≥1 post-vaccination relapse. However, further data were not available until the second interim analysis. Within a median follow-up of 4.5 months after the first vaccination, 9.3% of the 2661 PwMS studied in the second follow-up had ≥1 post-vaccination relapse [24]. Extrapolation of the relapse rate to one year after the initial vaccination resulted in an ARR of 0.19. In parallel, the ARR of a historical reference cohort of matched, unvaccinated PwMS from the German MS Registry from 2020 (before the approval of SARS-CoV-2 vaccines) was calculated to be 0.15. Since no pre-vaccination relapse rate could be calculated in the second interim evaluation, another reference cohort of SARS-CoV-2-vaccinated PwMS from the German MS Registry was examined for changes in relapse rate before and after vaccination. This showed no significant change in ARR (pre-vaccination 0.109 vs. post-vaccination 0.116) [24]. The major advantages of the third follow-up were the longer median observation time since the first vaccination of 1.2 years and the temporal recording of post-vaccination relapses (time from vaccination to subsequent relapse) as well as the request for physician-diagnosed relapses, which significantly increased the power of the results compared to the previous evaluations. Although the proportion of patients reporting ≥1 relapse in the year after initial vaccination is higher in the new analysis (13.8%) than in the previous analyses, information on the temporal distribution of relapses is now available. Thus, relapses occurred significantly faster after the first vaccination than after subsequent vaccinations (median: 4 weeks vs. 9.5 and 13.5 weeks, respectively), but most relapses occurred after the second or booster vaccination. An indication that post-vaccination relapse activity is not unusually high is provided by the evolution of the ARR over successive analyses. At the third follow-up, the ARR was significantly lower than the extrapolated ARR in the previous study [24], with minimal overlap in the confidence intervals (0.15 [0.14–0.17] vs. 0.19 [0.17–0.21]). Thus, the ARR with more than one year of follow-up is almost similar to the ARR of the historical pre-vaccination reference cohort of PwMS from the German MS Registry from the previous interim analysis [24]. However, it is not possible to directly compare these two study cohorts due to the different data collection methods in the observational study (patient-reported) and the German MS Registry (physician-reported in clinical or established MS centers).

Other population-based studies have also examined post-vaccination relapse activity. In a questionnaire-based cohort study of 2261 DMT-treated and SARS-CoV-2-vaccinated PwMS (over 80% with mRNA-based vaccines) from 16 Polish MS centers, post-vaccination relapses occurred in only 99 patients (4.4%) [44]. The majority of these 99 relapsing patients received the first or second vaccination more than 3 weeks before relapse. Thus, compared to our study, post-vaccination relapses occurred much less frequently, which further supports the overall safety of SARS-CoV-2 vaccines. However, all Polish patients received DMT, whereas only about three-quarters of our PwMS patients received DMT. Kong et al. studied 556 patients with neuromyelitis optica spectrum disorder (NMOSD) and 280 PwMS over a median follow-up of 9.4 months for post-vaccination disease activity compared to propensity score-matched patients without prior vaccination against SARS-CoV-2, with the majority of patients being unvaccinated (N = 649) [45]. Relapses were proportionally more frequent in unvaccinated PwMS than in vaccinated PwMS (7.7% vs. 5.1%), with no significant sociodemographic or clinical differences between the two groups. The analysis showed no differences in the risk of relapse for both PwMS and NMOSD patients [45]. To contextualize the results, it should be noted that the vaccines used in the study by Kong et al. were inactivated SARS-CoV-2 vaccines, not mRNA- or vector-based vaccines as in our study. Despite studies indicating that SARS-CoV-2 vaccines appear to be safe for PwMS or patients with other autoimmune diseases [44,45,46,47], there are numerous case reports of PwMS suffering from post-vaccination relapses [48,49]. Case reports are important tools to draw attention to these phenomena, but causalities between vaccination against SARS-CoV-2 and subsequent MS relapses often cannot be elucidated. Therefore, further studies with larger study populations based on longitudinal data were recommended and may also give information about the risk of other immune-mediated side effects in PwMS.

Instead of attributing relapses exclusively to the previous SARS-CoV-2 vaccination, previous SARS-CoV-2 infections may also provide an explanation. Several mechanisms have been discussed that may lead to the development of MS and other neurological diseases in the course of SARS-CoV-2 infection or to an exacerbation of MS [50,51,52]. Cytokines and chemokines are modulated during SARS-CoV-2 infection and have the potential to influence glial cell interactions in the development of MS [52]. Risk factors associated with the occurrence of post-vaccination relapses in our study were female sex (vs. male), RRMS course (vs. SPMS), autoimmune comorbidities (vs. none), relapse activity in the year before the first vaccination (vs. none), age ≤ 40 years (vs. >40 years), and lack of DMT (vs. DMT). These risk factors suggest that the health and therapeutic status of PwMS before and during initial vaccination may play a greater role in the development of post-vaccination relapses than SARS-CoV-2 vaccination per se. Patients in whom disease activity was under control prior to vaccination, e.g., by appropriate DMT, have a significantly lower risk of post-vaccination relapse than patients in an active phase of MS or without DMT. The relationship between age, gender, and the occurrence of MS relapses is complex. Various studies suggest that as individuals age, the risk of MS relapse tends to decrease and that older individuals may have a more stable disease course [53]. Gender-based differences such as gene expression or sex hormones are apparent, as MS more commonly affects women than men [12,13]. However, there may be individual differences in the frequency of relapses, and not all PwMS experience relapses to the same degree. It is essential to highlight that these observations are rooted in overall trends, and personal experiences can vary. To our best knowledge, there is currently no evidence indicating that age or gender have a significant impact on the occurrence of relapses explicitly following SARS-CoV-2 vaccination in PwMS [40,44,45,54,55]. Our findings suggest that relapse occurrence is more likely to be independent of SARS-CoV-2 vaccination, with younger women having more post-vaccination relapses than men in the present analysis. However, in older age groups, post-vaccination relapse rates were comparable between men and women.

This 1-year update of the vaccine observational study has some limitations. First, there was no adequate method for us to compare ARRs before and after vaccination. To calculate the ARR in the year prior to the initial vaccination, data on the number of relapses during that year were needed. In our study, however, we only had data on the last relapse prior to the first vaccination. On the basis of patient-reported data collection by questionnaire, it can never be completely excluded that pseudo-relapses, e.g., in the form of Uhthoff phenomena [56], were recorded in addition to clinical MS relapses. To minimize this risk factor, we asked explicitly for physician-diagnosed MS relapses and their timing in the third follow-up survey. However, even assuming that some of the reported relapses were actually relapse-like phenomena, this would result in a reduced ARR. This reinforces the statement that the vaccines appear to be safe with respect to subsequent MS relapses. Rather, the results suggest that treatment and disease activity status at the time of vaccination may be the more important factors in the occurrence of relapses. In addition, a major advantage of the third follow-up is the long observation period of more than one year since the first vaccination. Thus, the limitation regarding over- or underestimation of the relapse rate due to seasonal relapse activity [57,58] in the case of a short observation period, which was expressed in the previous evaluation, could also be eliminated.

## 5. Conclusions

In conclusion, this 1-year update of the prospective, non-interventional safety and tolerability study of SARS-CoV-2 vaccinations showed that most PwMS analyzed (86.2%) did not experience post-vaccination relapses. Among the patients who reported post-vaccination relapses, most developed relapses after the second or booster vaccination, respectively. The shortest time between the vaccination and subsequent relapses was after the initial vaccination. The ARR of the study population with a median follow-up period of 1.2 years since the first vaccination was 0.15, thus substantially lower than the ARR of 0.19 at the interim analysis with a median observation period of 4.5 months. In addition, PwMS with female gender, RRMS course, coincident autoimmune diseases, relapses within the year prior to the first vaccination, an age between 18 and 40 years, and no DMT at baseline had significantly higher ARRs than patients with male gender, SPMS course, no autoimmune comorbidities, no pre-vaccination relapse activity, an age >40 years, and DMT at baseline, respectively. Thus, it appears that post-vaccination relapses are less related to previous SARS-CoV-2 vaccinations than to the disease progression and treatment status before or at the time of vaccination. In particular, age, disease course, disability level, DMT status, and relapse activity within the year prior to the first vaccination seem to play a role in risk assessment. To ensure successful vaccination for PwMS, it is recommended to first stabilize their disease levels. Using appropriate DMTs for this purpose is essential, as certain DMTs have been found to lead to weakened post-vaccination immune responses and diminished vaccination efficacy. Additionally, optimizing the timing of vaccination can minimize the risk of relapse and improve vaccination impact. For example, when using anti-CD20 antibodies, vaccination should be scheduled towards the end of the therapy cycle. It is also important to assess vaccination success serologically [59]. Future population-based studies should also focus on the timely and quantitative relationship between SARS-CoV-2 infections and subsequent MS exacerbations, particularly with regard to relapses, central nervous system lesions, new-onset or worsened MS symptoms, and long-lasting infection symptoms (Long-COVID, Post-COVID).

## Figures and Tables

**Figure 1 vaccines-11-01859-f001:**
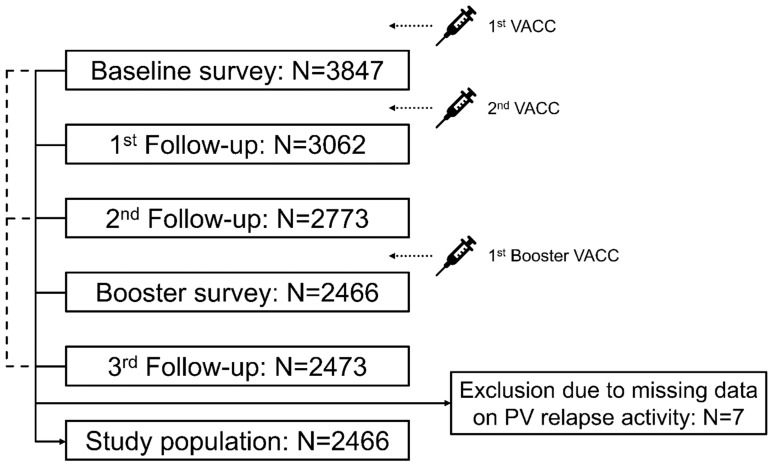
Study design. Five patient-reported online surveys were conducted within the year after PwMS received their first VACC: after the first and prior to the second VACC (baseline survey), immediately after the second VACC (first follow-up), approximately three months after the second VACC/completed basic immunization (second follow-up), after the first booster VACC (booster survey), and approximately one year after the first VACC (third follow-up). Inclusion criteria for this analysis were an age of ≥18 years, MS diagnosis, ≥1 VACC, and provision of data regarding the presence or absence of PV relapses in the third follow-up. Most PwMS who participated in the third follow-up also took part in the other four surveys (N = 2204). Of the remaining participants in the third follow-up, 187 attended the baseline survey as well as follow-ups 1 and 2, 66 participated in the baseline survey in the second follow-up as well as the booster survey, and 16 attended the baseline survey and follow-up 2. Seven of the 2473 participants in the third follow-up were excluded from this analysis due to missing data on PV relapse activity. N—number of patients; PV—post-vaccination; PwMS—people with multiple sclerosis; SARS-CoV-2—severe acute respiratory syndrome coronavirus 2; VACC—SARS-CoV-2 vaccination.

**Figure 2 vaccines-11-01859-f002:**
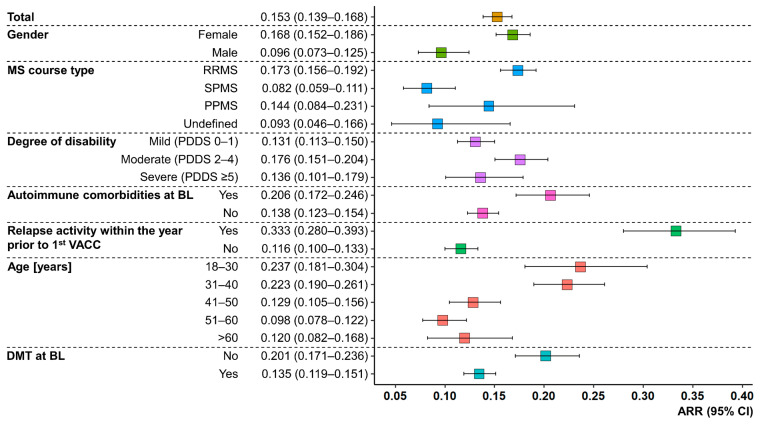
Relapse activity after any SARS-CoV-2 vaccination in people with MS. Colored boxes represent the ARRs of PwMS, stratified by age, gender, MS course type, degree of disability, DMT status at BL, relapse activity within the year prior to the first VACC, and the presence/absence of coincident autoimmune diseases at BL. Whiskers symbolize 95% CIs. Age ≤ 40 years, female gender, RRMS course, no DMT at BL, relapses within the year before the first VACC, and autoimmune comorbidities at BL resulted in substantially higher ARRs compared to patients with age > 40 years, male gender, SPMS course, DMT at BL, no relapses within the year prior to the first VACC, and absent autoimmune comorbidities at BL. ARR—annualized relapse rate; BL—baseline; CI—confidence interval; DMT—disease-modifying therapy; MS—multiple sclerosis; PDDS—patient-determined disease steps; PPMS—primary progressive MS; PwMS—people with MS; RRMS—relapsing-remitting MS; SARS-CoV-2—severe acute respiratory syndrome coronavirus 2; SPMS—secondary progressive MS; VACC—vaccination against SARS-CoV-2.

**Figure 3 vaccines-11-01859-f003:**
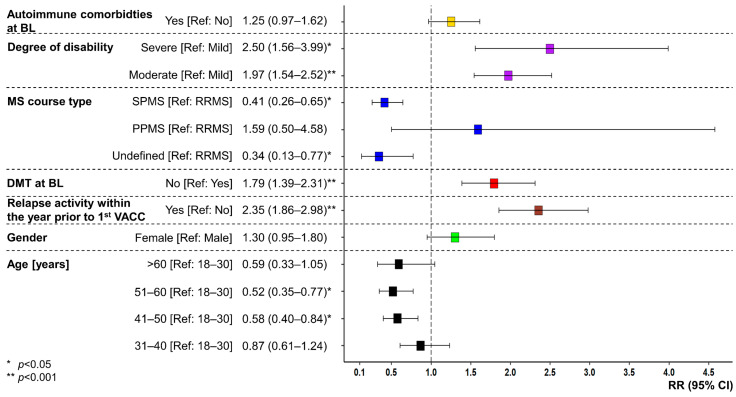
Risk factors for relapses following the vaccination against SARS-CoV-2 in people with MS. RRs (colored boxes) and 95% CIs (whiskers) for the occurrence of PV relapses were calculated using a multivariable negative binomial model. Higher risk of PV relapses was found for moderate to severe disability levels, no DMT at baseline, and relapses within the year before the first VACC. Lower risk of PV relapses was significantly associated with older age, SPMS, and undefined MS course. BL—baseline; CI—confidence interval; DMT—disease-modifying therapy; MS—multiple sclerosis; *p*—*p*-value; PPMS—primary progressive MS; Ref—reference; RR—risk ratio; RRMS—relapsing-remitting MS; SARS-CoV-2—severe acute respiratory syndrome coronavirus 2; SPMS—secondary progressive MS; VACC—vaccination against SARS-CoV-2.

**Table 1 vaccines-11-01859-t001:** Study population stratified by relapse activity following SARS-CoV-2 vaccination.

	Total (N = 2466)	PV Relapse (N = 341)	No PV Relapse (N = 2125)	*p*
**Gender**, N (%)				**0.001 ^Fi^**
Female	1939 (78.6)	294 (86.2)	1645 (77.4)	
Male	521 (21.1)	47 (13.8)	474 (22.3)	
Divers	6 (0.2)	0 (0.0)	6 (0.3)	
**Age [years] at BL**, median (Q25, Q75) *	46.9 (37.8, 54.8)	41.4 (34.1, 51.4)	47.8 (38.6, 55.2)	**<0.001 ^U^**
**Age groups [years]**, N (%)				**<0.001 ^Chi^**
18–30	226 (9.2)	44 (13.0)	182 (8.6)	
31–40	615 (25.0)	120 (35.5)	495 (23.4)	
41–50	691 (28.1)	88 (26.0)	603 (28.5)	
51–60	693 (28.2)	63 (18.6)	630 (29.7)	
>60	232 (9.4)	23 (6.8)	209 (9.9)	
**MS disease course at BL**, N (%)				**<0.001 ^Chi^**
RRMS	1827 (74.1)	285 (83.6)	1542 (72.6)	
SPMS	434 (17.6)	33 (9.7)	401 (18.9)	
PPMS	102 (4.1)	12 (3.5)	90 (4.2)	
Undefined	103 (4.2)	11 (3.2)	92 (4.3)	
**Disability level (PDDS) at FU3**, N (%) *				**0.025 ^Chi^**
Mild (0–1)	1248 (52.3)	153 (47.7)	1095 (53.1)	
Moderate (2–4)	829 (34.8)	133 (41.4)	696 (33.7)	
Severe (≥5)	307 (12.9)	35 (10.9)	272 (13.2)	
**Coincident autoimmune diseases at BL**, N (%)	530 (21.5)	95 (27.9)	435 (20.5)	**0.003 ^Chi^**
**DMT at BL**, N (%)	1806 (73.2)	225 (66.0)	1581 (74.4)	**0.001 ^Chi^**
IFNβ/GLAT	535 (21.7)	59 (17.3)	476 (22.4)	**0.090 ^Fi^**
CLAD/DMF/TER	507 (20.6)	77 (22.6)	430 (20.2)	
S1P RM	306 (12.4)	28 (8.2)	278 (13.1)	
anti-CD20 MAB	283 (11.5)	42 (12.3)	241 (11.3)	
Natalizumab	102 (4.1)	10 (2.9)	92 (4.3)	
Other	65 (2.6)	9 (2.6)	56 (2.6)	
Unknown DMT	8 (0.3)	0 (0.0)	8 (0.4)	
**Relapse within the year prior to 1st VACC**, N (%) *	362 (20.0)	91 (34.7)	271 (17.5)	**<0.001 ^Chi^**
**Relapse within 6 months prior to 1st VACC**, N (%) *	197 (10.9)	57 (21.8)	140 (9.0)	**<0.001 ^Chi^**
**Relapse within 3 months prior to 1st VACC**, N (%) *	87 (4.8)	29 (11.1)	58 (3.7)	**<0.001 ^Chi^**
**Time from last relapse (before 1st VACC) to 1st VACC [years]**, median (Q25, Q75)	3.2 (1.3, 6.9)	1.85 (0.6, 4.5)	3.5 (1.5, 7.2)	**<0.001 ^U^**

Anti-CD20 MAB—anti-CD20 monoclonal antibody (ocrelizumab/ofatumumab/rituximab); BL—baseline survey; Chi—chi-square test of independence; CLAD—cladribine; DMF—dimethyl fumarate; DMT—disease-modifying therapy; Fi—Fisher’s exact test; FU3—third follow-up; GLAT—glatiramer acetate; IFNβ—interferon beta (interferon beta-1a/interferon beta-1b/peginterferon beta-1a); MS—multiple sclerosis; N—number of patients; PDDS—patient-determined disease steps; PV—post-vaccination; Q25—25% quantile; Q75—75% quantile; RRMS—relapsing remitting MS; S1P RM—sphingosin-1-phosphate receptor modulator (fingolimod/ozanimod/siponimod); SARS-CoV-2—severe acute respiratory syndrome coronavirus 2; SPMS—secondary progressive MS; TER—teriflunomide; U—Mann-Whitney U test; VACC—SARS-CoV-2 vaccination; *—denominators may differ due to missing values.

**Table 2 vaccines-11-01859-t002:** Vaccination scheme of MS patients stratified by relapse activity following SARS-CoV-2 vaccination.

	Total (N = 2466)	PV relapse (N = 341)	No PV relapse (N = 2125)	*p*
**1st VACC**, N (%) *				0.675 ^Fi^
Ad26.COV2.S	26 (1.1)	6 (1.8)	20 (0.9)	
AZD1222	265 (10.8)	38 (11.1)	227 (10.7)	
Elasomeran	241 (9.8)	33 (9.7)	208 (9.8)	
NVX-CoV2373	3 (0.1)	0 (0.0)	3 (0.1)	
Tozinameran	1926 (78.3)	264 (77.4)	1662 (78.4)	
**2nd VACC**, N (%) *				0.359 ^Fi^
Ad26.COV2.S	1 (<0.1)	0 (0.0)	1 (<0.1)	
AZD1222	77 (3.2)	5 (1.5)	72 (3.5)	
Elasomeran	284 (11.9)	41 (12.5)	243 (11.8)	
NVX-CoV2373	1 (<0.1)	0 (0.0)	1 (<0.1)	
Tozinameran	2024 (84.8)	283 (86.0)	1741 (84.6)	
**1st Booster VACC**, N (%) *				0.610 ^Fi^
Ad26.COV2.S	5 (0.2)	1 (0.3)	4 (0.2)	
AZD1222	7 (0.3)	1 (0.3)	6 (0.3)	
Elasomeran	728 (33.2)	92 (32.2)	636 (33.3)	
NVX-CoV2373	1 (<0.1)	1 (0.3)	3 (0.2)	
Tozinameran	1453 (66.2)	191 (66.8)	1262 (66.0)	

MS—multiple sclerosis; N—number of patients; PV—post-vaccination; SARS-CoV-2—severe acute respiratory syndrome coronavirus 2; VACC—SARS-CoV-2 vaccination; *—denominators may differ due to missing values.

## Data Availability

Anonymized data will be made available on request by any qualified investigator under the terms of the registries’ usage and access guidelines and subject to the informed consent of the patients.

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
