# Peer review of "Occurrence and Risk Factors of Relapse Activity after Vaccination against COVID-19 in People with Multiple Sclerosis: 1-Year Follow-Up Results from a Nationwide Longitudinal Observational Study"

_vaccines, 2023, doi:10.3390/vaccines11121859_

Round 1
Reviewer 1 Report
Comments and Suggestions for Authors
Dear colleagues,
Thank you very much for your kind invitation to review the article “Relapse activity after vaccination against COVID-19 in people with multiple sclerosis: 1-year follow-up results from a nationwide longitudinal observational study”.
In the article colleagues have introduced current data. It is now known that vaccination can provoke the development of autoimmune inflammation. The effect of COVID-19 vaccine on autoimmune inflammation is currently limited.
The following items should be corrected:
- The aim of the study is not fully understandable and consistent with the title of the article – as it should be defined straightforwardly, otherwise upon review of the whole article it is still not clear what conclusions might be expected.
- In chapter Materials and methods should present the characteristics of autoimmune diseases.
- It is necessary to specify what autoimmune diseases were determined and by what methods?
- Did colleagues use a validated questionnaire?
- Please include in conclusions recommendations for the practical community.
Author Response
Dear colleagues,
Thank you very much for your kind invitation to review the article “Relapse activity after vaccination against COVID-19 in people with multiple sclerosis: 1-year follow-up results from a nationwide longitudinal observational study”.
In the article colleagues have introduced current data. It is now known that vaccination can provoke the development of autoimmune inflammation. The effect of COVID-19 vaccine on autoimmune inflammation is currently limited.
The following items should be corrected:
- The aim of the study is not fully understandable and consistent with the title of the article – as it should be defined straightforwardly, otherwise upon review of the whole article it is still not clear what conclusions might be expected.#
Reply: We thank the reviewer for this valuable comment. The title and the description of the study objectives were revised (pages 2–3, lines 96–101).
- In chapter Materials and methods should present the characteristics of autoimmune diseases.
Reply: We have added some information on coincident autoimmune diseases in the methods section (page 3, lines 122–124). However, these data were collected during the baseline survey, which was described in detail in the supplement of a previous article (reference 24: Frahm, N.; Fneish, F.; Ellenberger, D.; Haas, J.; Löbermann, M.; Peters, M.; Pöhlau, D.; Röper, A.-L.; Schilling, S.; Stahmann, A.; et al. Frequency and Predictors of Relapses Following SARS-CoV-2 Vaccination in Patients with Multiple Sclerosis: In-terim Results from a Longitudinal Observational Study. J Clin Med 2023, 12, 3640, doi:10.3390/jcm12113640). We have also added this reference (page 3, lines 127–129).
- It is necessary to specify what autoimmune diseases were determined and by what methods?
Reply: For a better understanding of the data collection, we have referred to a previous article that describes the baseline and the first two follow-up questionnaires in detail (reference 24: Frahm, N.; Fneish, F.; Ellenberger, D.; Haas, J.; Löbermann, M.; Peters, M.; Pöhlau, D.; Röper, A.-L.; Schilling, S.; Stahmann, A.; et al. Frequency and Predictors of Relapses Following SARS-CoV-2 Vaccination in Patients with Multiple Sclerosis: In-terim Results from a Longitudinal Observational Study. J Clin Med 2023, 12, 3640, doi:10.3390/jcm12113640). In addition, we have added the questionnaire of the third follow-up as Supplementary Document S1 to the methods section (page 3, lines 129–133).
- Did colleagues use a validated questionnaire?
Reply: The questionnaires were collaboratively developed by the German MS Registry and specialists from its scientific advisory board and the German MS Society (DMSG) in consultation with the United Kingdom MS Registry and the MS Data Alliance. After development, several rounds of testing were conducted by employees of the MS Registry and DMSG as well as PwMS (page 3, lines 106–110). The questionnaires comprehensively surveyed MS clinical situations and vaccine-related occurrences. The baseline and first two follow-up questionnaires have been provided as supplementary material in a previous article (reference 24: Frahm, N.; Fneish, F.; Ellenberger, D.; Haas, J.; Löbermann, M.; Peters, M.; Pöhlau, D.; Röper, A.-L.; Schilling, S.; Stahmann, A.; et al. Frequency and Predictors of Relapses Following SARS-CoV-2 Vaccination in Patients with Multiple Sclerosis: Interim Results from a Longitudinal Observational Study. J Clin Med 2023, 12, 3640, doi:10.3390/jcm12113640). In order to ensure completeness, we have included the questionnaire for the third follow-up survey (1-year data) as Supplementary Document S1 in this manuscript (page 3, lines 129–133).
- Please include in conclusions recommendations for the practical community.
Reply: We thank the reviewer for raising this suggestion. We have extended our findings' implications for clinical practice (page 14, lines 473–479).
Kind regards
Niklas Frahm
Reviewer 2 Report
Comments and Suggestions for Authors
This manuscript reports findings from a German nationwide study on multiple sclerosis (MS) patients who received COVID-19 vaccinations. Of the 2466 participants, 13.8% experienced post-vaccination relapses, mainly after the second or booster dose. Younger age, female gender, severe disability, autoimmune diseases, relapsing-remitting MS, lack of disease-modifying therapy, and recent pre-vaccination relapses were identified as risk factors. The study suggests that patients' health status before and during vaccination may influence relapse occurrence more than the vaccination itself. Overall, most MS patients did not experience post-vaccination relapses, emphasizing the importance of considering individual health factors in risk assessment.
Although the manuscript is well-written, I recommend some minor revisions to enhance its clarity and overall quality.
1. Remove the extra Figure 3 on Page 7.
2. In the Discussion section, expand the information about the correlation between gender, age, and relapse occurrences. Consider rearranging Tables 1 and 2 or adding an extra table specifically showing the correlation based on gender and age.
3. Provide a more detailed explanation of the methods used for data analysis. Clarify the choice of non-parametric tests, particularly addressing why non-parametric tests were favored over parametric tests, considering the large sample size.
4. Justify the use of non-parametric tests despite the larger sample size. Explain why parametric tests were not chosen, considering the usual preference for their greater power and accuracy with large samples.
5. Explain the rationale behind choosing Fisher's exact test for analysis. Address concerns raised by researchers about its application, particularly in the context of sample size.
6. Address how the significant difference in sample sizes between male and female groups was accounted for during the analysis. Provide details on any adjustments made or considerations taken into account.
7. Discuss whether there was a way to incorporate the history of COVID-19 disease into the analysis, as mentioned in the Discussion section. If feasible, provide insights into how this additional factor was considered or controlled for in the study.
Author Response
This manuscript reports findings from a German nationwide study on multiple sclerosis (MS) patients who received COVID-19 vaccinations. Of the 2466 participants, 13.8% experienced post-vaccination relapses, mainly after the second or booster dose. Younger age, female gender, severe disability, autoimmune diseases, relapsing-remitting MS, lack of disease-modifying therapy, and recent pre-vaccination relapses were identified as risk factors. The study suggests that patients' health status before and during vaccination may influence relapse occurrence more than the vaccination itself. Overall, most MS patients did not experience post-vaccination relapses, emphasizing the importance of considering individual health factors in risk assessment.
Although the manuscript is well-written, I recommend some minor revisions to enhance its clarity and overall quality.
- Remove the extra Figure 3 on Page 7.
Reply: We thank the reviewer for this suggestion. The multi-panel Figure 3 was split into Supplementary Figure S3 (univariable negative binomial model) and Figure 3 (multivariable negative binomial model).
- In the Discussion section, expand the information about the correlation between gender, age, and relapse occurrences. Consider rearranging Tables 1 and 2 or adding an extra table specifically showing the correlation based on gender and age.
Reply: We thank the reviewer for these suggestions. The discussion has been expanded regarding the correlation of age, gender and relapse occurrence (page 13, lines 423–436) and we added the variable “Age groups” to Table 1. In addition, we have created Supplementary Table S1 characterizing MS patients stratified according to the occurrence of post-vaccination relapses, age and gender (pages 4–5, lines 199–203).
- Provide a more detailed explanation of the methods used for data analysis. Clarify the choice of non-parametric tests, particularly addressing why non-parametric tests were favored over parametric tests, considering the large sample size.
Reply: Non-parametric tests are generally more robust in situations where parametric assumptions like normality are not met, especially with large sample sizes. Parametric tests may become sensitive to deviations from normality in such cases. Of course, data transformation followed by parametric tests would be an alternative, however there are many possible situations where no transformation can lead to normality. Thus, non-parametric approaches may be preferred (pages 3–4, lines 146–154). This is also hinted by the references below:
Reference 28: Fneish, F., Ellenberger, D., Frahm, N. et al. Application of Statistical Methods for Central Statistical Monitoring and Implementations on the German Multiple Sclerosis Registry. Ther Innov Regul Sci 57, 1217–1228 (2023). https://doi.org/10.1007/s43441-023-00550-0
Reference 29: Nahm FS. Nonparametric statistical tests for the continuous data: the basic concept and the practical use. Korean J Anesthesiol. 2016 Feb;69(1):8-14. doi: 10.4097/kjae.2016.69.1.8. Epub 2016 Jan 28. PMID: 26885295; PMCID: PMC4754273.
Reference 30: Mishra P, Pandey CM, Singh U, Keshri A, Sabaretnam M. Selection of appropriate statistical methods for data analysis. Ann Card Anaesth. 2019 Jul-Sep;22(3):297-301. doi: 10.4103/aca.ACA_248_18. PMID: 31274493; PMCID: PMC6639881.
- Justify the use of non-parametric tests despite the larger sample size. Explain why parametric tests were not chosen, considering the usual preference for their greater power and accuracy with large samples.
Reply: When the assumptions of parametric tests are not met, even though we do have a large sample, the non-parametric tests are the best option. Non-parametric tests do not assume a specific shape for the population distribution. In situations where the true distribution is unknown or complex, non-parametric tests provide a more flexible approach. Non-parametric tests also allow for hypothesis testing and group comparisons without assuming the specific parameters of the population distribution. This is particularly useful when dealing with real-world data that may not fit to theoretical expectations. Moreover, non-parametric tests are less effected by extreme values, outliers and skewed distributions.
- Explain the rationale behind choosing Fisher's exact test for analysis. Address concerns raised by researchers about its application, particularly in the context of sample size.
Reply: We thank the reviewer for this valuable comment. Fisher’s exact test is one alternative in dealing with categorical endpoints when chi-square test assumptions are not. Another common alternative would be the chi-square test with Yates Continuity correction. Researchers have investigated and discussed the situation and suitability for each test and other possible tests (references below). However, we used Fisher’s exact test since Yates corrections has been reported to be conservative in controlling Type I error. Some studies also report Fisher’s exact test to also be overly conservative for large sample sizes however it remains more suitable when one of the cell frequencies is less than five (page 4, lines 153–160).
Reference 31 Paek, I. (2010). Conservativeness in Rejection of the Null Hypothesis When Using the Continuity Correction in the MH Chi-Square Test in DIF Applications. Applied Psychological Measurement, 34(7), 539–548. doi:10.1177/0146621610378288
Reference 32 Crans GG, Shuster JJ. How conservative is Fisher's exact test? A quantitative evaluation of the two-sample comparative binomial trial. Stat Med. 2008 Aug 15;27(18):3598-611. doi: 10.1002/sim.3221. PMID: 18338319.
Reference 33 Ludbrook J, Dudley H. Issues in biomedical statistics: analysing 2 x 2 tables of frequencies. Aust N Z J Surg. 1994 Nov;64(11):780-7. doi: 10.1111/j.1445-2197.1994.tb04539.x. PMID: 7945088.
Reference 34 Prescott RJ. Two-tailed significance tests for 2 × 2 contingency tables: What is the alternative? Stat Med. 2019 Sep 30;38(22):4264-4269. doi: 10.1002/sim.8294. Epub 2019 Jul 1. PMID: 31264237.
Reference 35 Kim HY. Statistical notes for clinical researchers: Chi-squared test and Fisher's exact test. Restor Dent Endod. 2017 May;42(2):152-155. doi: 10.5395/rde.2017.42.2.152. Epub 2017 Mar 30. PMID: 28503482; PMCID: PMC5426219.
- Address how the significant difference in sample sizes between male and female groups was accounted for during the analysis. Provide details on any adjustments made or considerations taken into account.
Reply: Since the objectives of the manuscript do not address gender-specific differences, we have not conducted any adjustment to address a specific size difference between both males and females. We would have conducted a further sensitivity analysis or propensity score matching or other matching techniques if we were focusing on the gender aspect. However, it is common to have such sample size variation in MS. The literature contains extensive information regarding the higher prevalence of MS in women compared to men (page 2, line 73–74).
Reference 12 Harbo HF, Gold R, Tintoré M. Sex and gender issues in multiple sclerosis. Ther Adv Neurol Disord. 2013 Jul;6(4):237-48. doi: 10.1177/1756285613488434. PMID: 23858327; PMCID: PMC3707353.
Reference 13 Coyle PK. What Can We Learn from Sex Differences in MS? J Pers Med. 2021 Oct 7;11(10):1006. doi: 10.3390/jpm11101006. PMID: 34683148; PMCID: PMC8537319
- Discuss whether there was a way to incorporate the history of COVID-19 disease into the analysis, as mentioned in the Discussion section. If feasible, provide insights into how this additional factor was considered or controlled for in the study.
Reply: We thank the reviewer for this valuable comment. The focus on SARS-CoV-2 infections preceding MS relapses represents a very interesting research perspective. Future studies could examine MS patients for the timing and frequency of subsequent relapses and prolonged infection symptoms after SARS-CoV-2 infections. This research objective was specified in the conclusion section (page 14, lines 479–483). However, its inclusion in the article may exceed its clarity and content capacity. We consider this aspect of the research to be the subject of a separate paper. To ensure clarity of the data collected, we have included Supplementary Document S1.
Kind regards
Niklas Frahm
Reviewer 3 Report
Comments and Suggestions for Authors
Comments here after
1. The abstract could be more succinct and clearer in presenting the study objectives, methodology, and key findings without overwhelming technical details.
2. The introduction covers a broad spectrum of information, but it could benefit from a more focused and streamlined approach, emphasizing the critical aspects relevant to the study's aim.
3. The extensive use of abbreviations (e.g., PV, VACC, MS) might hinder comprehension for readers unfamiliar with the field. Consider balancing abbreviations with full-term explanations or a glossary.
4. The article lacks a clear explanation of the specific questions or measures used to collect data on relapses after vaccination. Detailing these methods would enhance the study's transparency.
5. Clarification on the distinction between physician-diagnosed relapses and patient-reported symptoms would improve the reliability and interpretation of the results.
6. Comparing the relapse rates after vaccination to pre-vaccination periods or to a non-vaccinated MS cohort could provide valuable context for the significance of observed effects.
7. While the article discusses risk factors for relapses, it could delve deeper into the mechanisms underlying these associations, offering insights into why certain variables predispose individuals to relapses post-vaccination.
8. Elaboration on why certain vaccination schemes or intervals were chosen could enhance the readers' understanding of the study's rationale.
9. The article's figures and tables are informative, but optimizing their layout and clarity could improve the ease of interpretation for readers.
10. The conclusion briefly summarizes findings but could expand on the implications of these results for clinical practice or future research directions related to COVID-19 vaccination in MS patients.
Comments on the Quality of English LanguageMinor editing of English language required
Author Response
- The abstract could be more succinct and clearer in presenting the study objectives, methodology, and key findings without overwhelming technical details.
Reply: We thank the reviewer for bringing this point up. We revised the abstract accordingly (page 1, lines 27–46).
- The introduction covers a broad spectrum of information, but it could benefit from a more focused and streamlined approach, emphasizing the critical aspects relevant to the study's aim.
Reply: We tried to shorten the introduction by summarizing some information. We also revised the objective section at the end of the introduction in order to present them in more detail (pages 2–3, lines 49–101).
- The extensive use of abbreviations (e.g., PV, VACC, MS) might hinder comprehension for readers unfamiliar with the field. Consider balancing abbreviations with full-term explanations or a glossary.
Reply: We reduced the use of abbreviations (PV, VACC) through the whole manuscript to increase the comprehension of the text for readers unfamiliar with the field of interest.
- The article lacks a clear explanation of the specific questions or measures used to collect data on relapses after vaccination. Detailing these methods would enhance the study's transparency.
Reply: The detailed questionnaire of the third follow-up has been added to the methods section (Supplementary Document S1; page 3, lines 129–134).
- Clarification on the distinction between physician-diagnosed relapses and patient-reported symptoms would improve the reliability and interpretation of the results.
Reply: Overall, all data from the surveys could be classified as patient-reported. In the third follow-up (1-year data), in contrast to the previous surveys, we explicitly asked about relapses diagnosed by a physician (including the date of diagnosis) (page 3, lines 133–134). With this measure, we wanted to reduce the reporting of pseudo-relapses and other phenomena and focus only on the "real" relapses that were verified by a physician. However, it can never be completely excluded that pseudo-relapses were recorded in addition to clinical MS relapses, as mentioned in the limitations section.
- Comparing the relapse rates after vaccination to pre-vaccination periods or to a non-vaccinated MS cohort could provide valuable context for the significance of observed effects.
Reply: We thank the reviewer for raising this issue. Unfortunately, there is no adequate method for us to compare relapse rates before and after vaccination. To calculate the annualized relapse rate (ARR) in the year prior to the initial vaccination, we need data on the number of relapses during that year. In our study, however, we only have data on the last relapse prior to the first vaccination (page 13, lines 437–441). Nevertheless, in a previous article we compared the ARRs of a reference cohort of PwMS from the German MS Registry before and after the initial vaccination against SARS-CoV-2 (reference 24: Frahm, N.; Fneish, F.; Ellenberger, D.; Haas, J.; Löbermann, M.; Peters, M.; Pöhlau, D.; Röper, A.-L.; Schilling, S.; Stahmann, A.; et al. Frequency and Predictors of Relapses Following SARS-CoV-2 Vaccination in Patients with Multiple Sclerosis: In-terim Results from a Longitudinal Observational Study. J Clin Med 2023, 12, 3640, doi:10.3390/jcm12113640). This analysis showed no significant difference in ARR before and after vaccination (before: 0.109 [95% CI: 0.084–0.138] vs. after: 0.116 [0.088–0.151]), as seen in the discussion section (page 12, lines 360–364).
- While the article discusses risk factors for relapses, it could delve deeper into the mechanisms underlying these associations, offering insights into why certain variables predispose individuals to relapses post-vaccination.
Reply: We have added the reviewer's points to the discussion (page 13, lines 423–436).
- Elaboration on why certain vaccination schemes or intervals were chosen could enhance the readers' understanding of the study's rationale.
Reply: We thank the reviewer for this suggestion. A paragraph on various factors and strategies for deciding on vaccination schedules was added to the beginning of the discussion (page 11, lines 319–332). However, we did not have own data on the reasons for specific vaccination schedules.
- The article's figures and tables are informative, but optimizing their layout and clarity could improve the ease of interpretation for readers.
Reply: We have tried to improve the readability of the Figures 2 and 3 by using for example new color schemes. In addition, Figure 3 has been split into two figures (univariable negative binomial model: Supplementary Figure S3; multivariable negative binomial model: Figure 3).
- The conclusion briefly summarizes findings but could expand on the implications of these results for clinical practice or future research directions related to COVID-19 vaccination in MS patients.
Reply: We appreciate the reviewer for bringing up this point. We have extended our findings' implications for clinical practice (page 14, lines 473–479).
Kind regards
Niklas Frahm